# Characterization of Indonesian Sugar Palm Bunch (*Arenga longipes* Mogea) Properties for Various Utilization Purposes

**Luthfi Hakim** [1,*], **Apri Heri Iswanto** [1], **Evalina Herawati** [1], **Ridwanti Batubara** [1], **Yunida Syafriani Lubis** [2] and **Erlina Nurul Aini** [3]

1    Department of Forest Product Technology, Faculty of Forestry, Universitas Sumatera Utara, Jalan Lingkar Kampus USU, Kampus 2 USU Bekala, Simalingkar A, Pancur Batu, Deli Serdang 20353, Indonesia; apri@usu.ac.id (A.H.I.); evalina@usu.ac.id (E.H.); ridwanti@usu.ac.id (R.B.)
2    Research Center for Applied Botany, National Research and Innovation Agency, Jalan Raya Jakarta-Bogor KM 46, Cibinong 16911, Indonesia; yuni030@brin.go.id
3    Reseach Center for Biomass and Bioproduct, National Research and Innovation Agency, Jalan Raya Jakarta-Bogor KM 46, Cibinong 16911, Indonesia; erli010@brin.go.id
*    Correspondence: luthfi@usu.ac.id

**Abstract:** Sugar palm bunch/SPB (*Arenga longipes* Mogea) waste is a lignocellulosic material derived from the harvest of sugar palm fruit (*kolang-kaling*). Therefore, this study aims to examine the anatomical, physical, chemical, and mechanical characteristics of SPB. The anatomical characterization results showed that SPB had two forms of fibrovascular bundles (FVBs), namely large and small. Based on morphology, SPB fibers had a length, diameter, average lumen diameter, and cell wall thickness of approximately $1346.42 \pm 415.71$ μm, $20.05 \pm 3.81$ μm, $11.82 \pm 2.95$ μm, and $4.12 \pm 1.08$ μm, respectively. FVB tissue of the sample had a fairly high density of 4–6 FVB per 4 mm$^2$ with a non-vascular area/total area ratio of 57.25%. The results showed that the sample contained $45.31 \pm 3.20\%$ cellulose, $23.21 \pm 3.73\%$ hemicellulose, $27.23 \pm 4.23\%$ lignin, and $1.39 \pm 0.32\%$ ash content. In addition, the extractive content that dissolved in hot water, cold water, ethanol-benzene, and 1% NaOH was $4.79 \pm 0.84\%$, $7.12 \pm 0.68\%$, $7.27 \pm 2.38\%$, and $29.81 \pm 3.78\%$, respectively. The GC–MS analysis results showed that *A. longipes* bunch contained several compounds, including carboxylic acid (tetradecanoic, octadecanoic), methoxy group (3-hydoxyphthalide), and palmitic acid. Meanwhile, the FTIR analysis showed the presence of OH groups with high intensity, which were identified as aromatic groups, as well as phenol groups recognized as lignin. Based on these results, characteristics of SPB were more suitable as raw materials for biomass energy, absorbent activated carbon, composite board, and surfactant.

**Keywords:** sugar palm bunch; *Arenga longipes*; fibrovascular bundles; anatomical; chemical; physical; mechanical; properties

## 1. Introduction

The Indonesian community forest program is an initiative aimed at fostering economic growth within communities and developing agroforestry systems based on sugar palm tree. In addition, the selection of sugar palm tree as a primary commodity is due to its versatility, with various parts serving different purposes. The Indonesian sugar palm, scientifically known as *Arenga longipes* Mogea, serves as an essential food crop commodity that produces sap and is widely cultivated by local communities, particularly in northern Sumatra [1,2]. The plant has also been reported to provide an array of benefits derived from various parts, including the roots, stems, leaves, frond, and sap [3–5]. Nira is the main product extracted from this plant by tapping the stalk and can be further processed into palm sugar or traditional wine [6,7]. Apart from sap, palm tree also produces fruits, recognized as *kolang-kaling* in the local language, which can be used as a food ingredient due to their high carbohydrate content. These fruits are often found in bunches hanging down from

the plant's tip between the fronds with a single bunch producing up to 100 fruits. Despite the potential of the empty bunch, which accumulates at a rate of 5–10 per tree and can be harvested once every month, their underutilization after frond harvesting often leads to waste production [8].

Sugar palm bunch (SPB) are a combination of fiber remaining after separating the fruits from the fresh variants [9]. The abundant availability of SPB, characterized by its biodegradability, non-toxic nature, and status as a natural fiber, makes it an interesting product for exploration via various studies. At present, several studies have explored SPB as raw materials, specifically focusing on the preparation and characterization of nanocrystalline cellulose (NCC). In addition, these reports showed that NCC derived from SPB exhibits nano-sized dimensions and possesses a very high crystallinity [10]. According to Farma et al. [9], SPB can be used as a carbon nanofiber membrane for supercapacitor materials with good quality. Although existing studies on the use of palm tree parts focus on the fiber contained in the various parts, Huzaifah et al. [11] reported that the chemical, physical, mechanical, and thermal characteristics of derived fiber have distinct properties influenced by geographical differences. Hakim et al. [12] also reported variations in the anatomical, physical, chemical, and mechanical characteristics of fibrovascular bundles (FVBs) within *A. longipes* midribs based on longitudinal and radial positions. However, there are no reports on the basic characteristics of SPB of *A. longipes*, showing the pressing need for exploration.

Several studies have focused on the use of SPB for the production of charcoal and activated carbon, examining the basic characteristics of the raw materials. Understanding the fundamental properties of the raw material is crucial, as it enables more precise and effective use. Therefore, this study aims to examine the anatomical, chemical, physical, and mechanical properties of SPB.

## 2. Materials and Methods

SPB (*Arenga longipes* Mogea) were collected from community plantation cultivated from the agroforestry system in Sei Limbat Village, Langkat Regency, North Sumatra. Furthermore, SPB were obtained from bunches of sugar palm aged approximately 8–10 years. The samples were hand-picked from the plant at a height of 5 cm from the main stem. Figure 1 shows the SPB of *A. longipes*.

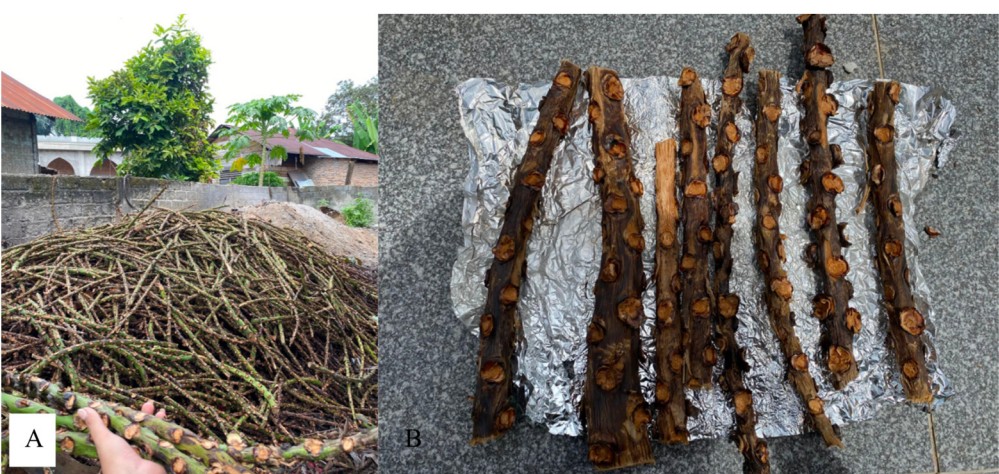

**Figure 1.** Photograph of SPB of *Areang longipes* Mogea. (**A**) Green of SPB; (**B**) air dry of SPB.

### 2.1. Sample Preparation

At 3 points in the cross-section, bunches were cut into sections of approximately 1–2 cm$^2$ in length. In addition, the cross-section (transverse section) and radial direction were the focus of the observations. The samples were then submersed in a boiling mixture of water and glycerin (volume ratio 1:10) for 2 h until full saturation to soften them. Transverse

and radial sections (10–15 μm thick) were cut from the block specimens using a sliding microtome (Reinchert, Baden-Baden, Germany), and safranin staining was used to clearly detect the area and structure of fibrovascular bundles.

### 2.2. Observation of Anatomical Properties

Observed anatomical properties of SPB included cell types, the location of fibers and vascular tissue, the number of sclerenchyma fibers and vascular tissue that filled the transverse section area, and the diameter of the fibrin–vascular bundle. Images from a light microscope (Olympus BX 51, Olympus Corporation, Tokyo, Japan) fitted with a digital camera (Olympus DP 70, Olympus Corporation, Tokyo, Japan) and imaging processing software system were used to observe the anatomical properties. In addition, transverse sectional data were used to calculate the frequency of FVB per 4 mm$^2$, followed by the determination of the ratio of the entire transverse sectional area to the vascular tissue area or sclerenchyma fiber.

### 2.3. Physical and Mechanical Properties Measurement of FVB

After air drying (moisture content: 8%–12%), FVB were tested for a total of 20 times. Approximately 90 mm long specimens were used, and FVB were attached to paper frames measuring 30 mm in gauge length using medium-viscosity epoxy adhesives (ALF Epoxy glue, P.T. Alfaglos, Semarang, Indonesia) in line with the ASTM D3379 preparation protocol [13]. Before being put through mechanical testing, the specimens were conditioned for one week at 20 °C and 60% relative humidity. The mechanical properties of FVB were assessed using a universal testing machine (Tensilon RTF 1350, Tokyo, Japan) with a crosshead speed of 1 mm/min in line with ASTM D882 standard [14]. Subsequently, the middle section of the supporting paper was clipped before testing.

### 2.4. Ultrastructure of SPB via Scanning Electron Microscopy

The changed FVB surface morphologies were assessed using scanning electron microscopy (SEM). After being sliced into 0.5 cm lengths, the samples were dried for an hour at 80 °C. To avoid specimen charging under the electron beam, the samples were coated with Pt–Pd and then scanned at an accelerating voltage of 1.5 kV. An SEM-JEOL-JSM-6390 (JEOL Ltd., Tokyo, Japan) was used to investigate FVB fibers.

### 2.5. Chemical Subtituents Analysis

Extractives were initially extracted using boiling water on the water bath equipment for 3 h, in line with ASTM D1110, from the 2 g oven-dried samples [15]. This step's weight loss was characterized as hot water extractive solubility. Extractive-free samples were then prepared according to ASTM D1105 standard [16]. Furthermore, the extractive of the 2 g oven-dried samples was extracted using a soxhlet extraction method for 4 h with an ethanol-toluene combination (ratio 1 L:427 mL). More analysis was needed to determine the amount of cellulose (ASTM D1103) [17], hemicellulose (ASTM D1104) [18], and klason lignin (ASTM D1106) [19] in these extractive-free samples. During the procedures, 3 duplicates of each analysis were performed.

### 2.6. Gas Chromatography–Mass Spectrometry (GC–MS) Analysis

The ethanol-benzene SPB extract from *Arenga longipes* was subjected to GC–MS analysis using an Agilent Technologist, Mass Hunter 5977 (GC–MS 7890B, Santa Clara, CA, United State of America) and a mass detector. A total of 1 milliliter per minute of helium was used as the carrier gas, and 1 L of the supernatant was injected into the GC–MS. Temperature increases in the GC–MS oven were carried out at rates of 15 °C/min from 80 °C to 200 °C, 5 °C/min from 280 °C to 280 °C, and a 5 min isothermal at 280 °C. A temperature of 230 °C was selected for the ion source, and 70 eV was chosen for the ionization voltage. The National Institute of Standards and Technology (NIST) database was used for GC–MS

interpretation. Subsequently, the unknown component's mass spectrum was compared to the spectrum of known components contained in the NIST library.

### 2.7. Fourier-Transform Infrared Spectroscopy (FTIR) Analysis

SPB powder samples were subjected to a 2 h boiling water soak to eliminate any extractives, followed by a 1 h conditioning period in room temperature water, a 12 h drying period at $35 \pm 5$ °C, and a 100-mesh screen powdering process using a 100-mesh screen. The FTIR-4200 spectrophotometer (8201PC-Shimadzu, Tokyo, Japan) was used for FTIR spectroscopy at room temperature (about 25 °C) with a 12 cm$^{-1}$ resolution using the KBr disk method to assign absorbance bands to distinct functional groups.

### 2.8. Thermogravimetry Analysis (TGA)

The changed FVB temperature-dependent thermal stability and the rate of mass change were determined via thermogravimetric examination using an Ekstar SIII-Type 7300 (Hitachi, Tokyo, Japan). To perform this study, materials weighing approximately 10 mg were heated in an alumina crucible from room temperature to 600 °C at a rate of 10 °C/min. Throughout the process, nitrogen was continuously pumped into the apparatus at a flow rate of 30 mL/min.

### 2.9. Index Crystallinity of SPB via X-ray Diffraction (XRD)

The crystallinity index (CI) was calculated by taking into account the crystalline and amorphous cellulose areas. Amorphous cellulose ($I_{am}$) was the lowest-intensity location (between 18° and 19°), while crystalline cellulose ($I_{002}$) was identified at a 2θ peak in the reflection plane position (highest intensity between 22.5° and 23°). Furthermore, radiation produced by the Maximax X-ray Diffractometer-7000 (Shimadzu, Kyoto, Japan) was used to obtain the diffraction spectra at room temperature (20–22 °C). The detector was positioned in the 2θ range, which spanned 5° to 70°, and the measurements were performed at 40 kV and 20 mA at a scan speed of 2°/min. Using Segal's procedure [20], the percentage crystallinity index (%CI) of the cellulose was determined by following equation:

$$\%CI = \frac{(I_{002} - I_{am})}{I_{002}} \times 100 \tag{1}$$

where $I_{002}$ is the maximal peak intensity at a 2θ angle of approximately 22–23°, and $I_{am}$ is the minimum peak intensity (amorphous region) at a 2θ angle of approximately 18–19°.

## 3. Results

### 3.1. Anatomical Properties

Figure 2 shows the anatomical characteristics of FVBs in *Arenga longipes*. The anatomy of the SPB was dominated by FVB tissue. According to previous studies, FVBs could be defined as tissue consisting of sclerenchyma fibers and vascular tissue bound into a complete bundle. All palm plants including the Indonesian sugar palm (*A. longipes*) contained FVB tissue. The tissue played an essential structural role through its fibers and transport function through the vessel. The perforation plate of the vessel in SPB of *A. longipes* was found to be of the scalariform type. Meanwhile, on the cell walls in the tracheary elements of the xylem, a spiral-shaped secondary wall thickening was observed.

Table 1 shows that the frequency of FVBs in SPB of *A. longipes* was between four and six FVBs per 4 mm$^2$. The total transverse area and vascular tissue area of FVBs were 0.08–0.55 mm$^2$ and 0.03–0.23 mm$^2$, respectively. Furthermore, the non-vascular tissue area was reported to be approximately 0.04 to 0.31 mm$^2$. Vascular tissue was the major parameter that differentiated FVBs and natural fibers (fiber bundle), where there was no vascular tissue found in natural fibers [21]. Vascular tissue had vessels that enhanced the porosity properties, while non-vascular tissue improved density [22]. The ratio of non-vascular tissue area of FVBs in *A. longipes* (54%–57%) was larger compared to the vascular tissue area (43%–46%). The results showed that the ratio of vascular tissue in

*A. longipes* was higher compared to *Salacca frond*. According to Hakim et al. [12], a value of 9.92%–12.08% was obtained in *Salacca frond*. Therefore, it was predicted that SPB of *A. longipes* could be more porous compared to *Salacca frond*. The good natural porosity could be beneficial for biomass as a carbon source, showing the potential of SPB to be a candidate for producing porous carbon with diverse applications, including adsorbent, catalyst, or electrode materials.

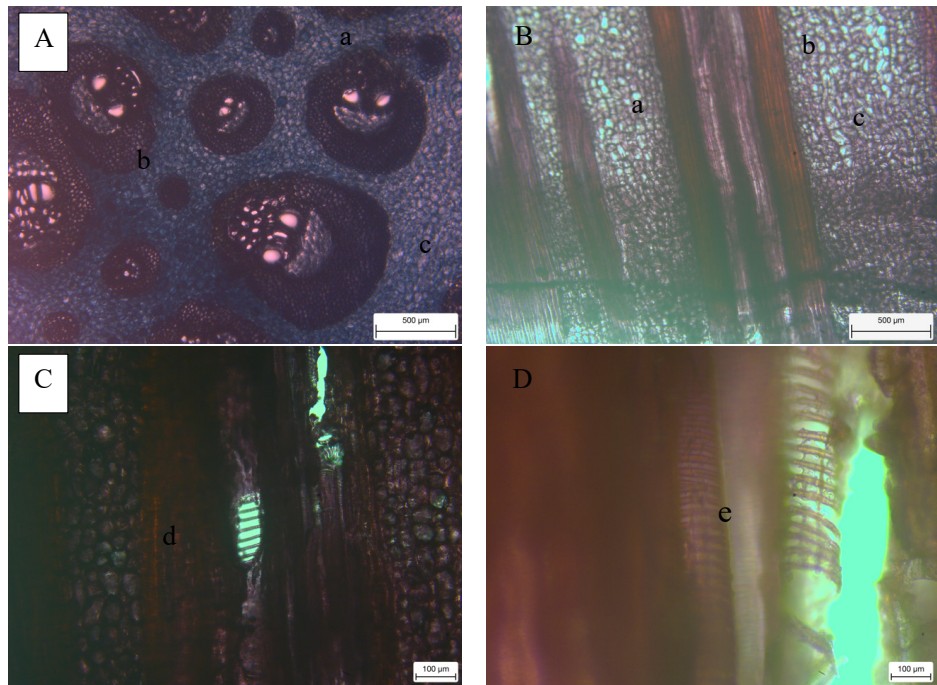

**Figure 2.** Light microscope photograph of the anatomical characteristics of SPB of *Arenga longipes*. (**A**) Cross-section; (**B**–**D**) radial section. Note: (a) Small FVB; (b) Parenchyma tissue; (c) Big FVB tissue; (d) Scalariform perforation plate; and (e) spiral-shaped secondary wall thickening.

**Table 1.** Anatomical characteristics of FVBs of SPB.

| Anatomical Characteristics | FVBs | |
|---|---|---|
| | **Small FVBs** | **Big FVBs** |
| Frequency of fibrovascular bundles per 4 mm$^2$ | 4–6 | 4–6 |
| Total transverse area (mm$^2$) | 0.08 ± 0.02 | 0.55 ± 0.14 |
| Vascular tissue area (mm$^2$) | 0.03 ± 0.01 | 0.23 ± 0.05 |
| Non-vascular tissue area (mm$^2$) | 0.04 ± 0.01 | 0.31 ± 0.10 |
| Ratio of vascular tissue area to total transverse area (%) | 46.30 | 42.74 |
| Ratio of non-vascular tissue area to total transverse area (%) | 53.69 | 57.25 |
| Ratio of vascular tissue area to non-vascular tissue area (%) | 86.00 | 74.00 |

### 3.2. Fiber Morphology and Fiber Derivates

Based on Figure 3, FVBs of SPB in *Arenga longipes* contained fiber, parenchyma tissue, vessel, and spiral-shaped secondary cell wall thickening. In line with previous studies, the spiral-type of cell wall thickening of the xylem was also clearly seen on the maceration result of FVBs. The fiber analysis of *A. longipes* SPB in this study is presented in Table 2. The results showed that the average fiber length, fiber diameter, lumen diameter, and cell wall thickness of SPB were 1.34 mm, 20.05 μm, 11.82 μm, and 4.12 μm, respectively. According to the International Association of Wood Anatomy (IAWA) classification, SPB fiber was categorized as moderate (0.9–1.6 mm) [23]. Compared to oil palm bunch and acacia fibers, the fiber length of SPB was longer. A previous study reported that oil palm bunch fiber

and acacia fibers had lengths of 0.77 mm and 0.84–0.86 mm, respectively [24]. The Runkle ratio of EFB fiber was 0.75, which was lower than 1. This showed that EFB fiber had a high lumen diameter with a thin wall, thereby making it easier to flatten [16]. The Muhlstep ratio and felting power of SPB fibers, namely 64.74 and 68.96, respectively, showed that its fiber was easy to flatten and could form inter-fiber bonding [23]. The coefficient of rigidity and flexibility ratio showed the flexibility degree of the fiber. In SPB *A. longipes*, the coefficient of rigidity and flexibility ratio were 0.37 and 0.54, respectively. This value was comparable to the coefficient of rigidity (0.25–0.31) and flexibility ratio (0.38–0.54) of bamboo fiber [25]. The results of the fiber morphology analysis showed that samples from SPB of *A. longipes* were suitable material for paper packaging.

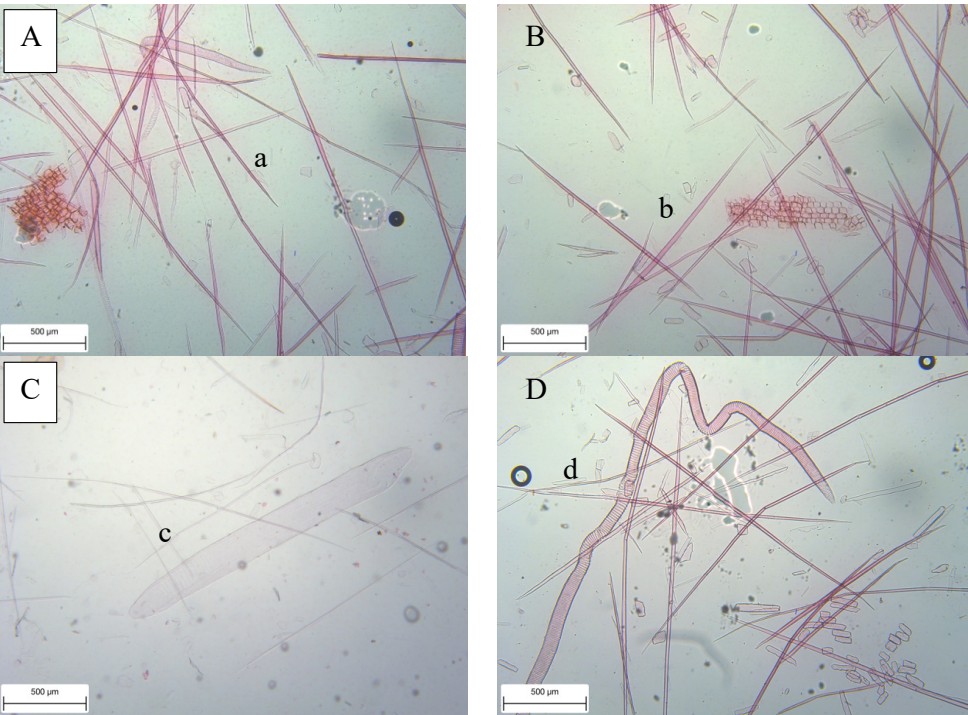

**Figure 3.** Light microscope photograph of FVB maceration. (**A**) fiber; (**B**) parenchyma; (**C**) vessel; (**D**) cell wall thickening. Note: (a) Fiber tissue; (b) parenchyma tissue; (c) vessel tissue; and (d) spiral-shaped secondary cell wall thickening.

**Table 2.** Fiber morphology and fiber derivates.

| Fiber Morphology | Value |
|---|---|
| - Fiber length (mm) | 1.34 ± 0.41 |
| - Fiber diameter (μm) | 20.05 ± 3.81 |
| - Lumen diameter (μm) | 11.82 ± 2.95 |
| - Cell wall thickness (μm) | 4.12 ± 1.08 |
| **Fiber Derivates** | |
| - Runkel ratio | 0.75 ± 0.33 |
| - Felting Power | 68.96 ± 22.81 |
| - Muhlstep ratio | 64.74 ± 9.22 |
| - Coefficient of rigidity | 0.37 ± 0.20 |
| - Flexibility ratio | 0.54 ± 0.15 |

### 3.3. Ultrastructure of FVB via Scanning Electron Microscopy (SEM)

The observation under SEM at cross-section for 25× magnification showed the varied size of FVBs, namely small and big sizes (Figure 4A,C). FVBs consisted of fibers and

other types of cells that constituted the sclerenchyma fibers, phloem, xylem vessels, and parenchyma. Xylem and phloem tissues were clearly distinguishable in FVBs. Based on the micrograph, SPB of *Arenga longipes* was very dense without any spaces, with a little parenchyma tissue. The results showed that it appeared to retain palm wood-like structural features. Despite having cell types of different sizes of FVBs (small and big FVBs), FVBs did not show any difference. The cells appeared like FVBs, which had a fiber structure and vascular tissue, and the differences were clearly visible. The scalariform perforation plate was also clearly visible at the borders between vessel tissue (Figure 4B). Zhu et al. [21] reported that FVBs on the Windmil palm (*Trachycarpus fortunei*) was divided into three categories based on the position of vascular tissue and sclerenchyma fiber. Similar results were obtained by Zhai et al. [22] regarding palm FVBs and Hakim et al. [12] concerning Salacca FVBs. FVB tissue of *A. longipes* SPB was vascular at the edge of the sclerenchyma fiber. Furthermore, it formed the spiral-shaped secondary cell wall thickening (Figure 4D). Based on results, there were no previous reports on this spiral-shaped secondary cell wall thickening in the anatomical structure of palm.

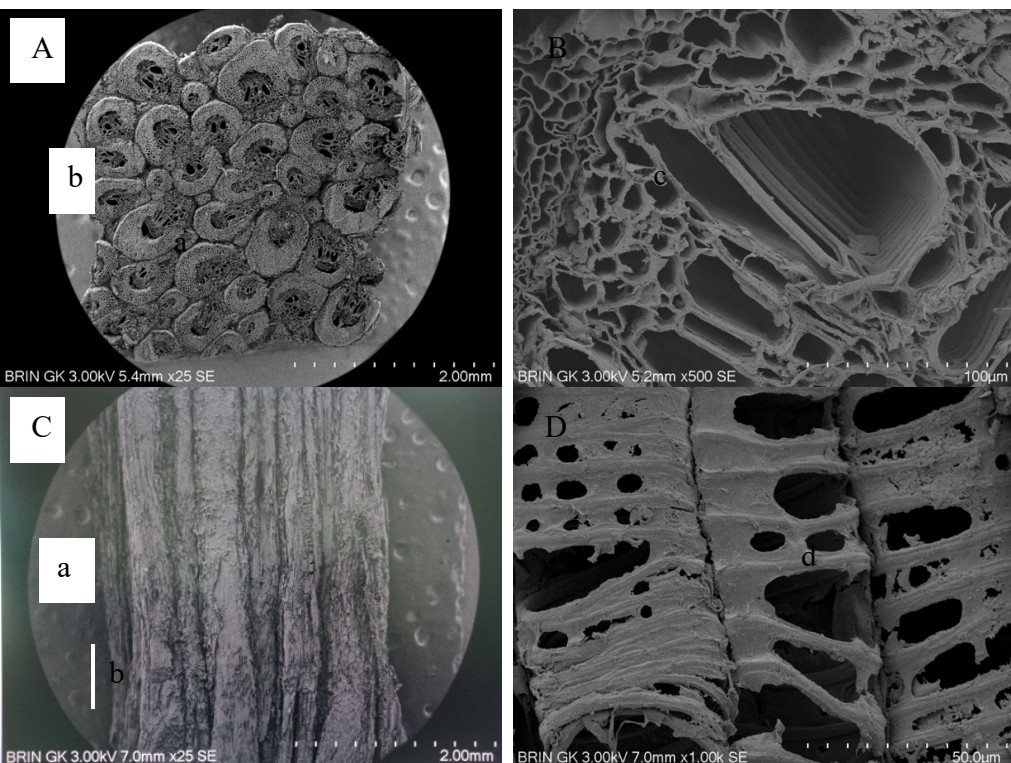

**Figure 4.** Micrograph of FVBs: (**A**,**B**) cross-section at 25× and 500×, respectively; and (**C**,**D**) radial section at 25× and 500×, respectively. Note: (a) Big FVB tissue; (b) small FVB tissue; (c) vessel tissue with scalariform perforation plate; and (d) spiral-shaped secondary cell wall thickening.

*3.4. Chemical Properties*

Table 3 shows the chemical properties of *Arenga longipes* bunch. Plant fibers were composite materials that had been intrinsically designed by nature. The majority of plant fibers consisted of cellulose, hemicellulose, lignin, extractives, and ash components. The chemical composition of palm fruit stalks was dominated by cellulose content of 45.3%, followed by 27.2% lignin, 23.21% hemicellulose, and 1.39% ash content. The highest extractive content in 1% NaOH soluble was 9.8%, followed by cold water- and hot water-soluble extractive content with values of 7.12% and 4.79%, respectively. When compared with a previous study conducted by Sahari et al. [26], the chemical composition of *A. longipes* bunch was slightly different from that of *A. pinnata*. The cellulose content and ash content of *A. longipes* bunch were lower compared to *A. pinnata* bunch at 61% and

3.4%, respectively. However, the lignin content and extractive content of *A. longipes* bunch were higher compared to *A. pinnata* bunch at 23.5% and 2.2%, respectively. The lignin and extractive content in *A. longipes* was disadvantageous for its use as pulp material. This was because the presence of lignin and extractive in high value inhibited the pulping process. When compared to other biomasses for pulp materials, the lignin content of SPB was lower than that of three bamboo species, including *B. vulgaris* (28.36%), *B. longispiculata* (27.45%), and *D. membranaceus* (29.02%) [27]. High extractive content could be considered advantageous for *A. longipes* bunch use as fuel material due to the high calorific value of biomass with high extractive content. The low ash content of *A. longipes* bunch also increased its desirability for fuel material. The chemical composition of plants varied significantly between plants and within different parts of the same plant. Due to the nature of growth, the percentage composition of each of these components varied depending on the prevailing conditions and several factors, such as soil quality, geographical location, weather conditions, age, and morphological part of the plant [28,29].

**Table 3.** Chemical properties.

| Chemical Properties | Value |
|---|---|
| Hot water solubility (%) | $4.79 \pm 0.84$ |
| Cold water solubility (%) | $7.12 \pm 0.68$ |
| 1% NaOH solubility (%) | $9.81 \pm 3.78$ |
| Ethanol-benzene solubility (%) | $7.27 \pm 2.38$ |
| Holocellulose (%) | $68.51 \pm 4.56$ |
| Cellulose (%) | $45.31 \pm 3.20$ |
| Hemicellulose (%) | $23.21 \pm 3.73$ |
| Klason lignin (%) | $27.23 \pm 4.23$ |
| Ash content (%) | $1.39 \pm 0.32$ |

*3.5. Physical and Mechanical Properties of FVB*

Table 4 shows the physical and mechanical properties of FVB of *Arenga longipes* bunch. The average diameter and density of FVBs of *A. longipes* were 0.042 mm and 0.42 g/cm$^3$, respectively. The average tensile strength and Young's modulus of FVBs were 357.79 MPa and 5.33 GPa, respectively. As density was known to affect fiber mechanical strength [30], the specific tensile strength and specific Young's modulus of *Arenga longipes* FVBs were also calculated with values of T 755.42 MPa and 12.69, respectively.

**Table 4.** Physical and mechanical properties of FVBs.

| Physical and Mechanical Properties | FVBs | |
|---|---|---|
| | Small FVBs | Big FVBs |
| Diameter (mm) | $0.047 \pm 0.14$ | $0.036 \pm 0.01$ |
| Density (g/cm$^3$) | $0.35 \pm 0.15$ | $0.48 \pm 0.11$ |
| Tensile strength (MPa) | $221.85 \pm 27.18$ | $493.72 \pm 24.92$ |
| Young's modulus (GPa) | $4.13 \pm 0.74$ | $6.52 \pm 0.66$ |
| Specific tensile strength (MPa) | $482.28 \pm 181.20$ | $1028.55 \pm 226.54$ |
| Specific Young's modulus (GPa) | $11.80 \pm 4.93$ | $13.58 \pm 6.01$ |

The physical properties of FVBs of *A. longipes* were similar to *Salacca frond* in Hakim et al.'s study [12]. The diameter and density of Salacca's FVBs were 0.036 mm and 0.35 g/cm$^3$ in *Salacca Sumatrana*, with 0.047 mm and 0.46 g/cm$^3$ in *Salacca zalacca*. Although their density was comparable, the mechanical properties of *A. longipes* FVBs, such as the tensile strength and Young's modulus, were higher compared to *Salacca frond* by 68%–85% and 76%–137%, respectively. *S. Sumatrana* and *S. zalacca* had tensile strength of 212.75 MPa and 193.51 MPa, with Young's modulus of 3.03 GPa and 2.25 GPa [12]. Compared to the strength of SPB from *A. pinnata*, the tensile strength of *A. longipes* was similar while its Young's modulus

was lower. According to Sahari et al. [26], tensile strength and Young's modulus of SPB *A. pinnata* were 365.1 MPa and 8.6 GPa, respectively. The good mechanical properties of FVBs from *A. longipes* were favorable for its use as particle board or paper packaging materials.

### 3.6. XRD (X-ray Diffraction) Analysis

For many years, XRD analysis has been used to identify the biomass structure, including crystalline regions, amorphous regions, and crystal shapes [31]. The result of XRD analysis of *Arenga longipes* bunch showed that the crystallinity index was 46.6%. Furthermore, cellulose was a straight-chain polysaccharide with three hydroxyl groups per glycoside molecule, which tended to create intramolecular or intermolecular glucose linkages [32]. The high cellulose content of *A. longipes* bunch was correlated with the high crystallinity index. This property contributed to the formation of the carbon structure in charcoal because the ideal pattern of charcoal structure typically had a relatively high degree of crystallinity.

The degree of cellulose crystallinity was a major factor in determining the potential of *A. longipes* bunch as a bioconversion feedstock. This affected the enzymatic hydrolysis process because low cellulose crystallinity increased the efficiency of cellulose hydrolysis via cellulase. The mechanical and thermal characteristics of a fiber could be influenced by the individual fibers' crystallinity. Figure 5 shows the diffractogram of *A. longipes* bunch. The pattern of X-ray diffraction for *A. longipes* bunch showed that the three strongest peaks were 16°, 22.2°, and 34.5°. Based on these results, the crystalline structure conformed to the structure of cellulose-I, as shown by a study by Qanytah et al. [33]. Furthermore, the crystal structure of *A. longipes* bunch was monoclinic, and this was consistent with Siruru et al. [32] on sago waste material, which was defined by the primary peak of the 2-theta diffraction angle between 15–17° and 21–22°.

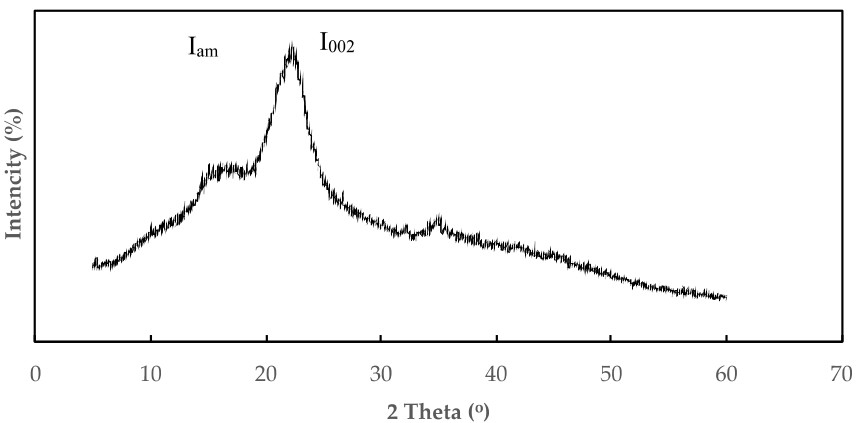

**Figure 5.** XRD diffractogram of SPB.

### 3.7. Gas Chromatography–Mass Spectrometry Analysis

The compounds generated in *Arenga longipes* bunch extractives were analyzed using GC–MS, and the results are presented in Table 5. Figure 6 shows the chromatograms corresponding to *A. longipes* bunch extractives. The peak area percentage of acids was approximately 61% with carboxylic acid being the major compound. Linear acids, such as tetradecanoic, hexadecenoic, and octadecanoic, were present with a significant area percentage ranging from 0.9% to 14.18%. Within the temperature range of 350–650 °C, long-chain carboxylic acids, including tetradecanoic and hexadecenoic acid, were found to be the main compounds. At high temperatures, the long-chain hydrocarbon part of the carboxylic acids cracked, leading to a noticeable increase in the generation of aromatics and cyclic hydrocarbons. The acid compounds were also identified in date palm biomass based on Benshidom et al. [34].

**Table 5.** The chemical compound of *A. longipes* bunch during GC–MS measurements.

| Peak | RT | Area (%) | Compound |
|---|---|---|---|
| 1 | 7.794 | 5.03 | 1-Dodecanamine, *N,N*-dimethyl- |
| 2 | 9.363 | 1.57 | 1-Tetradecanamine, *N,N*-dimethyl- |
| 3 | 9.79 | 2.45 | Tetradecanoic acid |
| 4 | 10.932 | 0.9 | Hexadecanoic acid, methyl ester |
| 5 | 11.186 | 14.18 | *n*-Hexadecanoic acid |
| 6 | 12.097 | 6.36 | 6-Octadecenoic acid, methyl ester, (Z)- |
| 7 | 12.247 | 1.61 | 1-Allyl-3-methylcyclohex-2-enol |
| 8 | 12.443 | 100 | *cis*-Vaccenic acid |
| 9 | 12.547 | 7.57 | Octadecanoic acid |
| 10 | 16.792 | 2.9 | Diisooctyl phthalate |
| 11 | 23.957 | - | 4-Butyl-*N*[(1*E*)-phenylmethylene]-aniline |
| 12 | 25.895 | - | Piperidine,1-[5-(1,3-benzodioxol-5-yl)-1-oxo-2,4-pentadienyl]- |
| 13 | 26.737 | - | 7-Dimethylocta-2,6-dien-1-yl palmitate |
| 14 | 28.94 | - | 3-hydoxyphthalide |
| 15 | 30.74 | - | Nonacosan-10-one |
| 16 | 31.651 | - | 2*S*-Chloro-2-(3′-methoxy-4′-hydroxybenzyl)-3*S*-(3″methoxy-4″-hydroxybenzyl)-.gamma.-butyrolactone |

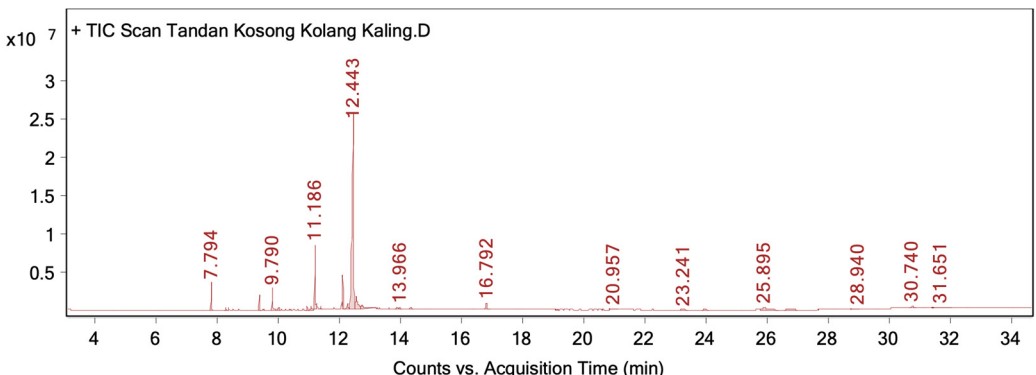

**Figure 6.** GC–MS spectrogram.

The decomposition products of lignin from *A. longipes* bunch were derivative with methoxy groups. Derivatives with 3-hydoxyphthalide (guaiacyl), also known as G-lignin, were found at a rating time of 28.94. The three types (S, G, and H) of lignin degradation products were identified in the pyrolyzates of oil palm biomass [35]. Some compounds of other types were also detected in the *A. longipes* bunch sample. Free fatty acids, such as 7-Dimethylocta-2,6-dien-1-yl palmitate (as known palmitic acid), were formed through the degradation of lipids. Palmitic acid, which was the major product of palm oil, was obtained, and this could be attributed to its high concentration of other (unspecified) chemicals as determined via compositional analysis [35].

*3.8. Thermogravimetry Analysis*

Figure 7 shows the TG/DTG and DTA graphs for *Arenga longipes* bunch during heating at a rate of 10 °C/min. The graphs provided insights into characteristics of lignocellulose materials during the heating process and the effect of heating rate and material type on their properties. The results showed that the mass loss of the sample material increased with increasing temperature. Previous studies on the pyrolysis of lignocellulosic biomass generally showed that the three main components, cellulose, hemicellulose, and lignin, were engaged in the primary thermal degradation process. Ishak et al. [36] stated that the thermal decomposition of sugar palm fiber consisted of four phases. The moisture evaporation was the first step, followed by the degradation of the lignocellulosic components of hemicelluloses, cellulose, lignin, and their ash.

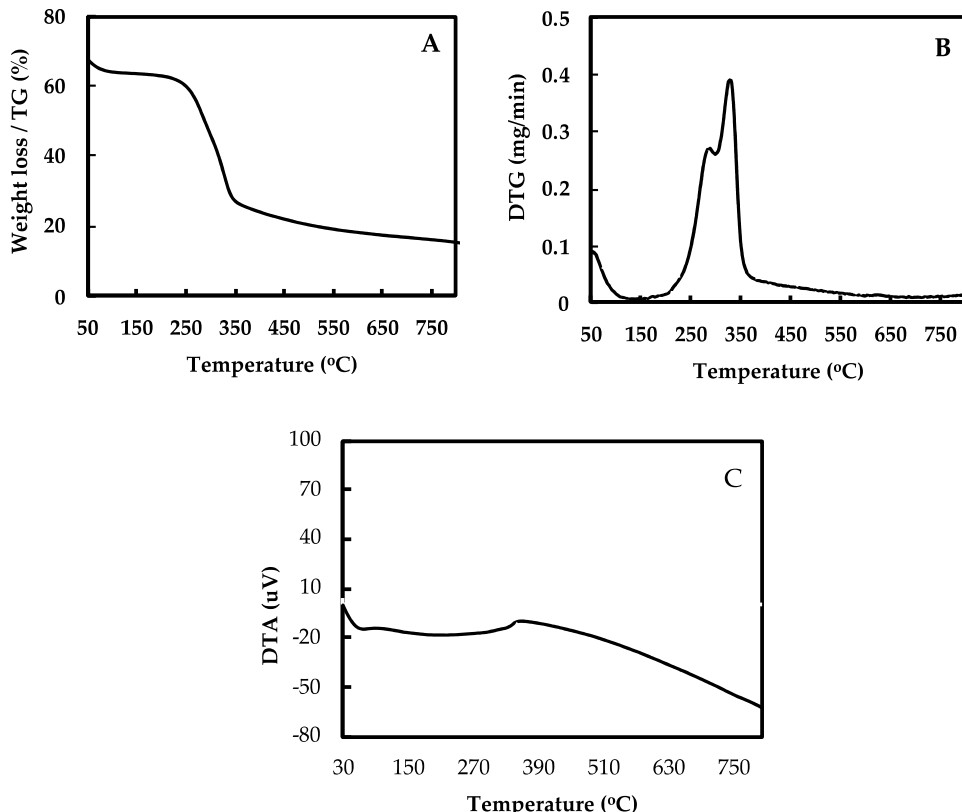

**Figure 7.** Thermogram of SPB. (**A**) Weight loss/TG; (**B**) DTG; and (**C**) DTA.

Based on the results depicted in Figure 7A, the initial phase comprised moisture evaporation absorbed in the *A. longipes* bunch, occurring within the temperature range of 50–150 °C and leading to a mass loss of 9.9%. This result was slightly different from Ishak et al. [36], in which the moisture evaporation in sugar palm fiber ranged from 45 to 123 °C. When the sample was heated, the initial mass of the sample decreased due to reduced bound water and volatile extractives, with volatile extractives likely moving to the fiber surface. According to the DTG curve (Figure 7B), the moisture evaporated from the sample was represented by the first peak. The next phase was hemicellulose decomposition at approximately 225 °C and completed at approximately 350 °C with a mass loss rate of 62%. The high mass loss percentage at this phase could be due to the cellulose and hemicellulose content of *A. longipes* bunch (Table 3). However, El-Sayed et al. [37] reported palm frond in this phase decomposition at 121–560 °C, and Ishak et al. [35] reported sugar palm fiber (from trunk) in the second phase decomposition of hemicellulose at 210–300 °C. From the results of this study, there was a potential that cellulose (amorphous cellulose) had been decomposed in this phase. This was because some studies reported that hemicellulose degraded at temperatures lower than 250 °C and cellulose degraded at temperatures between 250 °C and 500 °C [38,39]. The sequence of lignocellulosic degradation started with hemicellulose because it consisted of various saccharides, which were removed from the main stem and degraded to volatiles at low temperatures [40]. Cellulose degradation was the third phase of decomposition with the temperature set at 350 °C. A few amounts of amorphous cellulose were degraded first, while the degradation at this temperature (350 °C) was referred to as crystalline cellulose degradation. This phenomenon was observed in the DTG of the sample (Figure 7), where degradation of crystalline cellulose occurred at 300–350 °C with the highest peak of decomposition at 327 °C (degradation rate 0.39 mg/min).

The last phase of the process was lignin decomposition. When the lignin had completely degraded, it led to a mass loss of 76%–79% at 800 °C. Due to its wide range of decomposition and very low mass loss rate, it was difficult to observe the peak of lignin by

observing its DTG curves. Lignin decomposed more slowly and at temperatures exceeding 500 °C due to its linkage with a hydroxyl phenolic group [37].

### 3.9. Fourier-Transform Infrared Analysis

The results of the FTIR analysis are depicted in Figure 8. Based on the results, similar peaks were observed between normal (non-extracted) SPB and extracted SPB. In both normal and extracted SPB, the peaks on wavenumbers 925, 1035, 1229, 1378, 1456, 1608, and 3331 cm$^{-1}$ were found. The peaks on 925 and 1035 cm$^{-1}$ represented the absorption of polysaccharides and C-O (C-H) stretching from the glucose ring [41]. Meanwhile, the peaks on 1229, 1378, and 1456 cm$^{-1}$ could be associated with the C-O carbonyl band in lignin and xylan units [42], C-H in $CH_2$ groups of cellulose and hemicellulose [43], and C-H in $CH_3$ groups of lignin [44], respectively. The peaks on 1608 and 3331 cm$^{-1}$ represented the stretching of an aromatic ring framework in the lignin [45], stretching vibration of the OH group in polysaccharides, and also inter- and intramolecular hydrogen bond vibrations in cellulose [46,47]. The FTIR spectra analysis showed that the primary constituents in SPB of *A. longipes*, including cellulose, hemicellulose, and lignin, almost did not change before and after extraction.

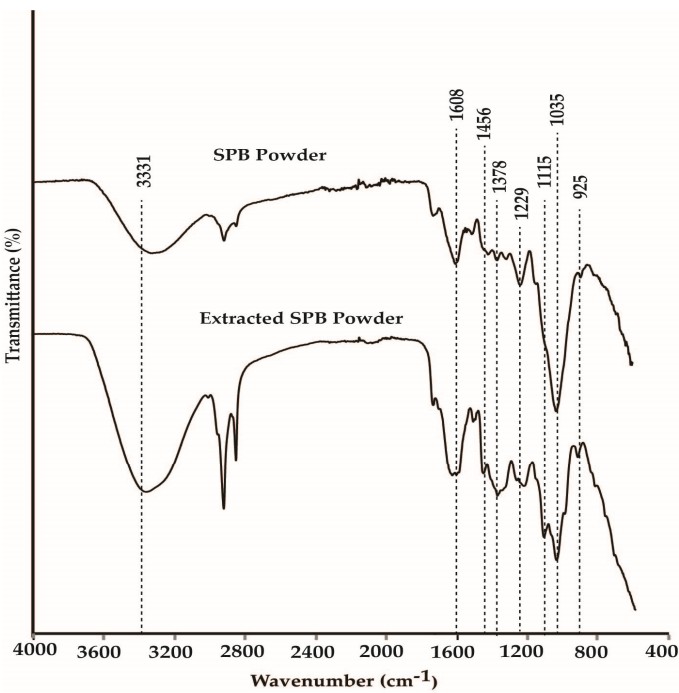

**Figure 8.** FTIR spectrogram of normal SPB and extracted ethanol-benzene SPB from *A. longipes*.

## 4. Discussion

Mogea [48] reported that the root, leaves, trunk, fruit, frond, bunch, and sap of sugar palm tree were often used for traditional products such as medicine, food packaging, construction, beverage, food, lightweight construction, sugar palm, or traditional wine, respectively. SPB of *Arenga longipes* were analyzed for its fundamental characteristics to determine the appropriate purposes of use according to its fundamental properties.

According to the anatomical properties and fiber morphology, SPB of *A. longipes* had FVB tissue with abundant sclerenchyma fibers. The sclerenchyma fibers had an abundance of lignocellulosic materials, which were used for pulp and paper, bio-composite products (particle board and fiberboard), charcoal, biomass energy, fermentation for bioethanol, etc. [49]. Furthermore, SPB of *A. longipes*, like other palm plants, contained FVB tissue, which mechanically had good tensile strength, Young's modulus values, and high cellulose crystallinity index. FVBs from monocot plants with good mechanical properties and high crystallinity index could be used as raw material for making excellent orientation

boards [50]. The physical properties of FVBs of *A. longipes* SPB showed the presence of low-density materials. The low density was advantageous as a raw material for making particle boards. This was because it facilitated the determination of the target density of the particle board as well as the transport and handling of raw materials [51]. The property also expedited the pyrolysis process in the carbonization and briquette dough-molding process [52].

Chemically, SPB of *A. longipes* had a high holocellulose constituent with relatively small solubility of extractive substances as well as high klason lignin content. These data were supported by the discovery of groups, showing the presence of polysaccharides and glucose rings in FTIR analysis. This had implications for the ease with which SPB material could be used as raw material for pulp and paper. The low extractive and lignin content made the pulping process easier and more efficient in the use of cooking chemicals for pulp [53]. The pulp- and paper-making processes could not be separated from the quality of the fiber raw materials. Previous studies showed that the fiber raw material was related to fiber morphology and dimension derivatives. Based on the morphology and derivative dimensions, SPB fiber was included in the intermediate category. This showed that SPB fiber from *A. longipes* could still be used as raw material for pulp and paper and fiberboard, but it had lesser quality compared to hardwood or softwood [54]. Furthermore, the GC–MS analysis showed that SPB of *A. longipes* had several acid contents, specifically canamine and decanoic acid. Decanoic acid was a saturated fatty acid with 10 carbons and was often present in palm kernel, coconut fat, and milk fat. Several studies showed that it could be used in the production of ammonium decanoate (a surfactant) and disperse phase [55]. Therefore, SPB of *A. longipes* had potential as raw material for the production of surfactant.

The thermal properties were identified using TGA, which was a powerful instrument for studying the devolatilization rate throughout the biomass combustion process and obtaining critical information for characterizing and understanding its behavior [56]. According to the results, SPB of *A. longipes* experienced gradual degradation up to a temperature of 350 °C, showing it had good thermal values. The weight loss of raw materials gradually decreased up to high temperatures of 700 °C. This showed that the thermal degradation of SPB of *A. longipes* was very good as a raw material for biomass energy [57].

In our conclusion, the various application purposes of SPB can be illustrated in Figure 9. The SPB of *A. longipes* have potential as raw material for manufacturing composite board, pulp, and paper; as activated carbon, surfactant, and biomass energy, such as pellet, charcoal, and briquette; and are absorbent.

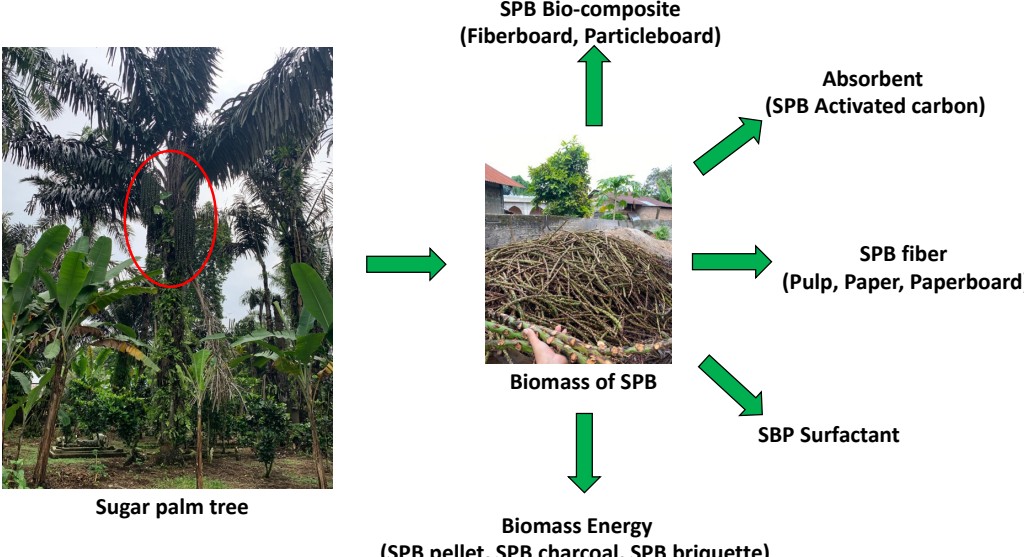

**Figure 9.** An illustration of various application purposes of SPB.

## 5. Conclusions

In conclusion, the properties of *Arenga longipes* Mogea bunch were characterized and analyzed in this study. The tensile strength of *A. longipes* was similar to *A. pinnata*, while its Young's modulus was lower. Furthermore, its crystallinity index was high due to its relatively high cellulose content. The results showed that *A. longipes* bunch was ideal to be used as raw material for charcoal products and composite boards due to the crystallinity index, which contributed to the formation of the carbon structure and mechanical properties of the board. GC–MS analysis showed that the *A. longipes* bunch contained several compounds, such as carboxylic acid, and methoxy group, which were derivatives from lignin degradation, as well as palmitic acid. The derivative compounds from cellulose decomposition were undetected in *A. longipes* bunch. Thermal properties of *A. longipes* bunch composition were found in four phases, where the sequence of degradation was moisture, hemicellulose, cellulose, lignin, and ash. Characteristics of *A. longipes* bunch depended on the part of the plant. Based on the optimum thermal properties, *A. longipes* bunch had the potential to be used as raw material for pulp, paper, charcoal products, briquette's absorbent activated carbon, bio-composite boards, and surfactants.

**Author Contributions:** All authors contributed equally to this work. All authors have read and agreed to the published version of the manuscript.

**Funding:** This study was funded by the Regular Fundamental Research Grant of the Directorate of Research, Technology, and Community Service, Ministry of Education, Culture, Research, and Technology, Republic of Indonesia No. 167/E5/PG.02.00.PL/2023 and Research Institution of Universitas Sumatera Utara No. 14/UN5.2.3.1/PPM/KP-DRTPM/B/2023.

**Data Availability Statement:** The data presented in this study are available on request from the corresponding author.

**Acknowledgments:** The authors are also grateful to the Laboratory of Forest Product Technology, Faculty of Forestry, Universitas Sumatera Utara, for providing equipment to perform the experiment.

**Conflicts of Interest:** The authors declare that there are no conflicts of interest.

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
