# Peer review of "Characterization of Indonesian Sugar Palm Bunch (Arenga longipes Mogea) Properties for Various Utilization Purposes"

_forests, doi:10.3390/f15020239_

Round 1

Reviewer 1 Report

Comments and Suggestions for Authors

The experimental article “Fundamental characteristics of Indonesian Sugar Palm Bunch (Arenga longipes Mogea) for utilization purposes” is devoted to the study of lignocellulosic raw materials – Indonesian Sugar Palm Bunch (Arenga longipes Mogea). The research carried out aims to characterize the properties of SPB which include anatomical, physical, chemical and mechanical properties. The authors compared the data obtained quite well with the results of other researchers. The article fully corresponds to the Forests publication. The authors are well versed in their topic and are ready to convey their results to readers with all the details. The article is written very well, clearly and without any doubt.

Flaws:

1. It would be nice to include a photograph of SPB in the “research object” section.

2. The drawing is difficult to read. Signatures need to be enlarged.

3. Determination of the chemical composition was also determined after the preparation described in paragraph 2.1? Now this is not clear. If so, this could affect the chemical composition.

Author Response

Dear Reviewer

We Really appreciate your efforts in handling our manuscript, entitled “Fundamental characteristic of Indonesian Sugar Palm Bunch (Arenga longipes Mogea) for utilization purposes”, with very constructive comments by the reviewers. The following are authors’ response to the reviewers’ comments: (All modifications in the text have been made in yellow highlight).

Reviewer 1:

The experimental article “Fundamental characteristics of Indonesian Sugar Palm Bunch (Arenga longipes Mogea) for various application purposes” is devoted to the study of lignocellulosic raw materials – Indonesian Sugar Palm Bunch (Arenga longipes Mogea). The research carried out aims to characterize the properties of SPB which include anatomical, physical, chemical and mechanical properties. The authors compared the data obtained quite well with the results of other researchers. The article fully corresponds to the Forests publication. The authors are well versed in their topic and are ready to convey their results to readers with all the details. The article is written very well, clearly and without any doubt.

Comments from Reviewer 1:

Comment from authors

A. Method

It would be nice to include a photograph of SPB in the “research object” section

Thank you for your suggestion, we have added the image of SPB in the method.

The drawing is difficult to read. Signatures need to be enlarged

Thank you for your feedback. We have improved the image quality so that it is clearer to read.

Determination of the chemical composition was also determined after the preparation described in paragraph 2.1? Now this is not clear. If so, this could affect the chemical composition.

The determination of the chemical composition was determined separately from preparation described in sub-title 2.1. The raw materials are natural and have not been contaminated with any chemicals

Reviewer 2 Report

Comments and Suggestions for Authors

Dear authors

This paer aimed to characterize the properties of sugar palm bunch including anatomical, physical, chamecal and mechanical properties. The organization, writing and presentation were accepted. However, the critical point is that the innovation of the paper is not well illustrated or not high. All methods used were commonly utilized, the difference is only the subject. Meanwhile, it is doubtful that if this paper fills into the scope of the Forests. It is suggested that comparing the properties of the SPB with tree, and other biomaterials.

In addition, some mistakes in details must be modified, especially the unit,   "cm2" should be "cm2", " Mpa" should be "MPa", as well as "Gpa". 

The tick labers of figures should be as large as the main text. Figure 5 and 6 must be modified.

The prospects of the SPB should be further discussed based on its unique properties compared with other materials.

Author Response

Dear Reviewer

We Really appreciate your efforts in handling our manuscript, entitled “Fundamental characteristic of Indonesian Sugar Palm Bunch (Arenga longipes Mogea) for utilization purposes”, with very constructive comments by the reviewers. The following are authors’ response to the reviewers’ comments: (All modifications in the text have been made in yellow highlight)

Reviewer 2:

This paper aimed to characterize the properties of sugar palm bunch including anatomical, physical, chemical and mechanical properties. The organization, writing and presentation were accepted. However, the critical point is that the innovation of the paper is not well illustrated or not high. All methods used were commonly utilized, the difference is only the subject. Meanwhile, it is doubtful that if this paper fills into the scope of the Forests. It is suggested that comparing the properties of the SPB with tree, and other biomaterials.

Comments from Reviewer 2:

Comment from authors

In addition, some mistakes in details must be modified, especially the unit, "cm2" should be "cm2", " Mpa" should be "MPa", as well as "Gpa". 

Thank you for your suggestion, we have revised some words.

The tick labers of figures should be as large as the main text. Figure 5 and 6 must be modified.

Thank you for your advice. We have improved the image quality so that it is clearer to read.

The prospects of the SPB should be further discussed based on its unique properties compared with other materials

Thank you for your suggestion, we have added the result and discussion in page 8, line 248.

Reviewer 3 Report

Comments and Suggestions for Authors

Authors performed a systematic study on the fundamental characteristics of Arenga longipes Mogea. This paper plays a good role in promoting the application of local agricultural products. Some details need to be revised for the better reading.

1. In the title, “fundamental characteristic” is too broad. It is suggested that the authors summarize the represented content in the order of research. For example, physical, chemical and mechanical properties. The composition also need to be included in the title.

2. In Figures 1 to 3, the mark is not clear. Please indicate the areas marked with letters with arrows or boxes.

3. The quality of Figures 4 to 7 is not sufficient to clearly convey the content, especially numbers or letters. Please adjust their format to be consistent.

4. Please provide a more detailed explanation of the purpose of utilization and provide pictures to illustrate.

Comments on the Quality of English Language

Moderate editing of English language is required.

Author Response

Dear Reviewer

We Really appreciate your efforts in handling our manuscript, entitled “Fundamental characteristic of Indonesian Sugar Palm Bunch (Arenga longipes Mogea) for utilization purposes”, with very constructive comments by the reviewers. The following are authors’ response to the reviewers’ comments: (All modifications in the text have been made in yellow highlight).

Reviewer 3:

Authors performed a systematic study on the fundamental characteristics of Arenga longipes Mogea. This paper plays a good role in promoting the application of local agricultural products. Some details need to be revised for the better reading.

Comments from Reviewer 3:

Comment from authors

A.    Title

In the title, “fundamental characteristic” is too broad. It is suggested that the authors summarize the represented content in the order of research. For example, physical, chemical and mechanical properties. The composition also need to be included in the title.

Thank you for your response. In our opinion, anatomy, fiber properties, physical properties, mechanics and chemical composition are fundamental characteristics that represent everything.

B.     Figures 1 to 3

The mark is not clear. Please indicate the areas marked with letters with arrows or boxes.

Thank you for your feedback. We have improved the image quality so that it is clearer to read.

C.     Figures 4 to 7

The quality of Figures 4 to 7 is not sufficient to clearly convey the content, especially numbers or letters. Please adjust their format to be consistent.

Thank you for your feedback. We have improved the image quality so that it is clearer to read.

Please provide a more detailed explanation of the purpose of utilization and provide pictures to illustrate.

Thank you for your feedback. We have improved the explanation in the last paragraph.

Round 2

Reviewer 2 Report

Comments and Suggestions for Authors

Dear authors

The pape has been much improved in presentation. Hope to see it be published. 

Reviewer 3 Report

Comments and Suggestions for Authors

Appropriate revisions were done by authors.